# UniGraph2: Learning a Unified Embedding Space to Bind Multimodal Graphs

## Abstract

Existing foundation models, such as CLIP, aim to learn a unified embedding space for multimodal data, enabling a wide range of downstream web-based applications like search, recommendation, and content classification. However, these models often overlook the inherent graph structures in multimodal datasets, where entities and their relationships are crucial. For example, in social networks, users are connected through friendships, follows, or interactions, and share content in various modalities like text and images. Multimodal graphs (MMGs) represent such graphs where each node is associated with features from different modalities, while the edges capture the relationships between these entities. On the other hand, existing graph foundation models primarily focus on text-attributed graphs (TAGs) and are not designed to handle the complexities of MMGs. To address these limitations, we propose UniGraph2[1], a novel cross-domain graph foundation model that enables general representation learning on MMGs, providing a unified embedding space. UniGraph2 employs modality-specific encoders alongside a graph neural network (GNN) to learn a unified low-dimensional embedding space that captures both the multimodal information and the underlying graph structure. We propose a new cross-domain multi-graph pre-training algorithm at scale to ensure effective transfer learning across diverse graph domains and modalities. Additionally, we introduce a new Mixture of Experts (MoE) component to align features from different domains and modalities, ensuring coherent and robust embeddings that unify the information across modalities. Extensive experiments on a variety of multimodal graph tasks demonstrate that UniGraph2 significantly outperforms state-of-the-art models in tasks such as representation learning, transfer learning, and multimodal generative tasks, offering a scalable and flexible solution for learning on MMGs.

## CCS Concepts

• **Information systems** → **Data mining**; *Social networks*; • **Computing methodologies** → **Neural networks**.

## Keywords

Graph Pre-Training, Graph Foundation Models, Web Mining, Multimodal Learning

---

[1]The code is available at https://anonymous.4open.science/r/UniGraph2/

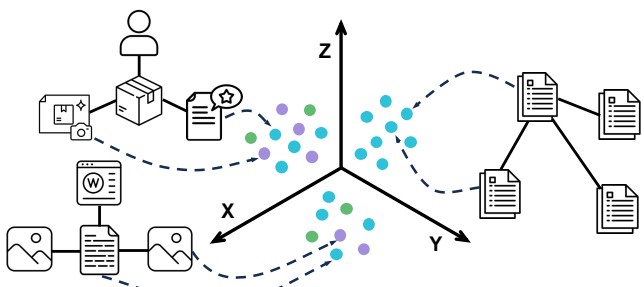

**Figure 1: UniGraph2 binds multimodal graphs from different graph domains to a unified embedding space, enabling diverse downstream tasks.**

**ACM Reference Format:**

Anonymous Author(s). 2018. UniGraph2: Learning a Unified Embedding Space to Bind Multimodal Graphs. In . ACM, New York, NY, USA, 12 pages. https://doi.org/XXXXXXX.XXXXXXX

**Relevance to the Web and to the track.** UniGraph2 is a large pre-trained model that processes and unifies multimodal graph data from various web domains, making it applicable to web-based applications like search, recommendation, and content classification.

## 1 Introduction

Real-world web applications increasingly rely on multimodal data, where information is derived from a variety of sources such as text, images, and audio [2, 34]. Recent foundation models have focused on learning a unified embedding space across different modalities that allows for the seamless integration of multimodal data, thereby enabling effective cross-modal interactions and supporting downstream applications [13, 37].

Models such as CLIP [37] have demonstrated the power of learning from multimodal data by mapping text and images into a shared embedding space. However, CLIP and similar models are fundamentally limited by their reliance on a 1-to-1 mapping between paired modalities, such as text-to-image alignment, ignoring more complex structures where nodes can be connected through many-to-many relationships and involve multiple modalities. These models fail to account for the graph structure present in numerous web domains, from social networks to e-commerce networks [9, 50, 56], where entities and their interactions are crucial to understanding the underlying relationships. For example, in e-commerce platforms, recommendation systems rely on complex networks of products, users, and their interactions [38]. Each node represents a a user or a product, and edges represent interactions like purchases, views, or reviews. Additionally, both users and products are associated with rich multimodal data: product descriptions (text), images (visual), and user reviews (text), and demonstration videos (audio and visual). Integrating these diverse data types within the graph structure is essential for accurate recommendations and personalized user experiences [11]. To address these challenges, Multimodal Graphs

(MMGs) have been introduced as a framework that combines graph structures with multimodal data [9, 56]. On MMGs, nodes are enriched with information from multiple modalities, allowing for a more comprehensive representation of entities and their relationships. However, existing MMGs learning methods can only train models individually for a specific graph and task [6, 50, 51], and cannot achieve cross-graph and cross-task transfer like foundation models do without retraining or fine-tuning.

Recently, there has been considerable progress in learning foundation models for text-attributed graphs (TAGs) [7, 15, 17, 19], which can be viewed as a special case of MMGs where the node features are are exclusively in the text modality. One prominent effort in this direction is UniGraph [19], which introduces a unified embedding space that combines graph structure and node-level textual information for all TAGs. UniGraph employs a masked prediction framework [14, 28, 36], inspired by the success of masked language models (MLMs) [28]. In this framework, UniGraph performs self-supervised pre-training by masking node-level text attributes and learning to predict the missing information based on the graph context. Despite its effectiveness on TAGs, UniGraph faces two significant limitations when extended to more complex settings. First, it is limited in its ability to generalize to MMGs, where nodes may contain features from diverse modalities such as images, in addition to text. Second, UniGraph focuses on pre-training on a single graph from one domain, which restricts its capacity to leverage knowledge across multiple domains. In training a foundation model, it is essential to employ more diverse pre-training data from different domains to enhance the model's generalization [1, 13, 37].

**Presented Work.** In this work, we propose UniGraph2, a graph foundation model for MMGs that provides a unified embedding space across graph domains and modalities, as shown in Figure 1. In UniGraph2, nodes are not restricted to textual attributes; instead, they can incorporate features from any combination of modalities. Similar to UniGraph, UniGraph2 adopts a masked prediction framework, but generalizes the masked prediction task to accommodate multimodal data. In this setup, the model is tasked with predicting missing node attributes, which could be text, image features, or any other modality, based on the graph structure and the available multimodal information. This allows the model to learn rich, unified representations that capture both the multimodal features of each node and the relationships encoded in the graph.

Furthermore, while UniGraph focuses on pre-training within a single graph domain, UniGraph2 introduces a more robust multi-graph pre-training strategy. In real-world applications, data often comes from multiple sources, each with different graph structures and node modalities. To handle this, UniGraph2 proposes a cross-domain multi-graph pre-training framework, which enables the model to learn compact and transferable knowledge across a diverse set of graph datasets with varying modality and domain distributions. A key component of this framework is the Mixture of Experts (MoE) [39], which is specifically designed to align node features from different domains and modalities. The MoE dynamically selects the most appropriate experts for each input data, ensuring that the diverse multimodal features are coherently integrated into the unified embedding space.

In summary, our key contributions in UniGraph2 are:

- We generalize the masked prediction framework used in Uni-Graph to support multimodal graphs, allowing nodes to include a variety of modalities such as text and images.
- We introduce a cross-domain multi-graph pre-training strategy, enabling UniGraph2 to learn unified and transferable representations across different graph domains and modalities.
- We incorporate an MoE component to align multimodal features from different graph domains, ensuring that the model dynamically selects and integrates the most compact and relevant information.
- We demonstrate through extensive experimentation that Uni-Graph2 outperforms state-of-the-art models in various multimodal graph learning tasks, including representation learning, transfer learning, and multimodal generative tasks, particularly when data is drawn from multiple graph domains.

## 2 Related Work

### 2.1 Multimodal Representation Learning

Building a general representation learning model for multimodal data has received significant attention in recent years, with various approaches aiming to unify learning across different modalities such as vision, language, and audio. Early approaches like Vision-Language Pre-training (VLP) models predominantly focus on learning from image-text data using contrastive learning and masked language modeling, leading to models such as CLIP [37] and ALIGN [26]. With the development of unified architectures [8, 25, 42] and pretraining tasks [3, 14, 28, 36], more work begin to explore effective alignment of representations for a wider range of different modalities, with the potential to expand to unlimited modalities [13, 47].

### 2.2 Multimodal Graph Learning

Most existing multimodal graph learning models primarily focus on knowledge graphs [6, 51] and natural sciences, such as molecular graphs [27] or brain graphs [46]. However, these models are specifically designed for particular tasks on individual graphs using domain knowledge and do not aim to learn a unified and general representation. They also cannot be transferred across different graphs, modalities, or tasks. Unlike these works, a recent work, MMGL [50] explores the use of foundation models from different modalities on MMGs, but it focuses solely on generative tasks.

### 2.3 Graph Foundation Models

Learning graph foundation models that can be transferred across different graphs [18, 19, 21, 35] and tasks [18, 20, 29] has recently received significant attention. Some works explore designing domain-specific graph foundation models, such as those for knowledge graphs [10] and molecular graphs [49]. Most existing research efforts are dedicated to using LLMs with strong generalization capabilities to solve graph learning tasks [17, 29, 40, 44]. However, how to effectively serialize graph data so that LLMs can understand the graph structure and graph learning tasks remains a barrier to further performance improvements [53]. Additionally, these models typically use the generative capabilities of LLMs to directly generate predicted labels, thus addressing representation learning tasks

on graphs. Due to the high computational cost, it is challenging to scale them to web-scale large graphs [17, 44].

## 3 Preliminaries

### 3.1 Multimodal Graphs (MMGs)

DEFINITION 1 (MULTIMODAL GRAPHS). *A Multimodal Graph (MMG) is defined as a graph $\mathcal{G} = (\mathcal{V}, \mathcal{E}, \mathcal{M}, \Omega)$, where $\mathcal{V}$ represents the set of nodes and $\mathcal{E}$ represents the set of edges. The function $\mathcal{M} : \mathcal{V} \to 2^{\Omega}$ maps each node $v \in \mathcal{V}$ to a subset of modalities $\Omega_v \subseteq \Omega$, where $\Omega$ denotes the set of all possible modalities, such as text, images, or other data types. Each node $v$ in $\mathcal{V}$ can possess multiple features from different modalities, but not all nodes are required to have features from every modality.*

For a Text-Attributed Graph $\mathcal{G}_{TAG} = (\mathcal{V}, \mathcal{E}, \mathcal{M}, \{text\})$, where each node has an associated text $t_v \in \mathcal{T}_{\mathcal{V}}$, we define the mapping function for MMGs as follows:

$$\mathcal{M}(v) = \{text\}, \text{ for all } v \in \mathcal{V}. \tag{1}$$

Here, $\Omega = \{text\}$ is the set of possible modalities, limited to textual data in this context.

### 3.2 General Representation Learning on MMGs

General representation learning [13, 33, 37, 47] on MMGs aims to learn a self-supervised pre-trained model that can infer meaningful representations for any new MMG, facilitating downstream tasks without the need for additional training or fine-tuning on new data.

PROBLEM 1 (GENERAL REPRESENTATION LEARNING ON MMGs). *Consider a collection of Multimodal Graphs (MMGs) in the pre-training set $\mathcal{D}_{pretrain}$, where each graph $\mathcal{G}_k = (\mathcal{V}_k, \mathcal{E}_k, \mathcal{M}_k)$ contains nodes $v_{ik} \in \mathcal{V}_k$ each associated with a set of modalities $\Omega_{v_{ik}} \subseteq \Omega$, encompassing various data types such as text, images, and other feature modalities. The challenge in general representation learning on MMGs involves self-supervised pre-training a function $f : \mathcal{V}_k \to \mathbb{R}^d$ across this diverse dataset. The objective is to develop a model that generalizes well to any new, unseen graph, enabling effective inference across various MMGs. For inference, the pre-trained model $f$ is applied to a new, unseen graph $\mathcal{G}^{inf} = (\mathcal{V}^{inf}, \mathcal{E}^{inf}, \mathcal{M}^{inf})$ to generate embeddings for its nodes, thereby facilitating downstream tasks on $\mathcal{G}^{inf}$ without further training.*

**UniGraph [19].** TAGs are a subset of MMGs where each node is associated with textual features. As a general representation learning model on TAGs, UniGraph unifies the learning process by integrating LM and GNN into a single encoder.

In UniGraph's pre-training, the masked prediction process can be mathematically formulated in two key steps:

(1) **Masked Encoding:** For each node $v \in \mathcal{V}$ has its textual feature $t_v$ partially masked and encoded by an LM $f_{\theta_1}^{LM}$, producing hidden representations $E_v = f_{\theta_1}^{LM}(\text{Mask}(t_v))$. The GNN $f_{\theta_2}^{GNN}$ propagates node embeddings across the graph, where the final node embedding is:

$$E'_{CLS} = f_{\theta_2}^{GNN}(\mathcal{G}_{TAG}, E_{CLS}), \tag{2}$$

with $E_{CLS}$ representing the embeddings of all nodes' [CLS] tokens from $f_{\theta_1}^{LM}$.

(2) **Decoding:** The MLP decoder $f_{\theta_3}^{Decoder}$ combines the masked textual embeddings $E_v$ and the graph embeddings $E'_{CLS}$ to reconstruct the masked tokens. The predicted probability distribution $P_v$ over the vocabulary is obtained via:

$$P_v = f_{\theta_3}^{Decoder}(\text{concat}(E_v, E'_{CLS})), \tag{3}$$

and the model minimizes the masked language modeling loss $\mathcal{L}_{MLM}$, formulated as:

$$\mathcal{L}_{MLM} = -\frac{1}{|\mathcal{V}|} \sum_{v \in \mathcal{V}} \sum_i I(v, i) \log P_v[i, T_i], \tag{4}$$

where $I(v, i)$ indicates masked positions and $T_i$ are the true tokens. The optimal parameters are obtained by:

$$\theta_1^*, \theta_2^*, \theta_3^* = \arg \min_{\theta_1, \theta_2, \theta_3} \mathcal{L}_{MLM}. \tag{5}$$

In inference, the pre-trained model is used to generate embeddings for any unseen TAG $\mathcal{G}_{TAG}^{inf} = (\mathcal{V}^{inf}, \mathcal{E}^{inf}, \mathcal{T}_{\mathcal{V}}^{inf})$ by processing the graph structure and node texts through the same encoder:

$$H^{inf} = f_{\theta_2^*}^{GNN}\left(\mathcal{G}_{TAG}^{inf}, X^{inf}\right), \text{ where } X^{inf} = f_{\theta_1^*}^{LM}(\mathcal{T}_{\mathcal{V}}^{inf}). \tag{6}$$

This process allows the model to generalize to new data, capturing both structural and textual graph attributes.

## 4 The UniGraph2 Framework

The overall framework of UniGraph2 is illustrated in Figure 2. The UniGraph2 framework introduces a unified approach to learning representations of multimodal graphs (MMGs), which consist of nodes with diverse modal features (such as text and images) and edges representing relationships between these entities. The framework comprises three key modules: the multimodal feature encoders, which process multimodal features through modality-specific encoders; the Mixture of Experts (MoE) module, which selects specialized MLP to align features across domains and modalities; and the decoders, which map the unified embeddings back into domain-specific inputs. The GNN operates as the central component that propagates node embeddings based on both their multimodal features and the underlying graph structure.

### 4.1 Multimodal Masking Strategies

In UniGraph2, masking strategies play a crucial role in the self-supervised learning framework for MMGs. The objective is to mask a portion of the node features and require the model to reconstruct them, thereby encouraging the model to effectively capture both the structural and multimodal information.

**Modality-Specific Encoding.** Before applying the masking process, modality-specific encoders are used to map raw data from different modalities into feature vectors. In the context of a multimodal graph $\mathcal{G} = (\mathcal{V}, \mathcal{E}, \mathcal{M}), \Omega$, where each node $v \in \mathcal{V}$ can have features from a subset of modalities $\Omega_v \subseteq \Omega$, the raw features are transformed using encoders specific to each modality (e.g., a language model for text, and a Vision Transformer for images).

Let $E_\omega$ represent the encoder for a modality $\omega \in \Omega$, and let $x_i^{(\omega)} \in \mathbb{R}^{d_{in}}$ denote the feature vector for node $v_i$ obtained from modality $\omega$. The modality-specific encoding can be expressed as:

$$x_i^{(\omega)} = E_\omega(v_i^{(\omega)}). \tag{7}$$

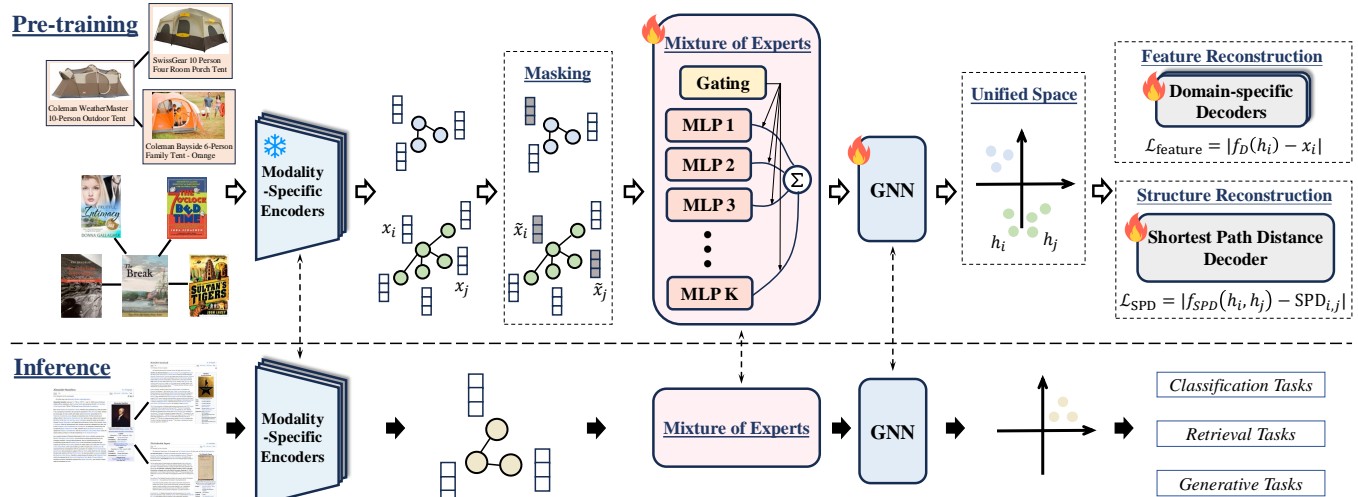

**Figure 2: Overview of the UniGraph2 framework. In pre-training, 1) UniGraph2 uses frozen Modality-Specific Encoders to encode raw multimodal data (e.g., text, images) into vector node features. Then, a portion of these node features is randomly masked. 2) Considering the diversity of node features across different modalities and graph domains, a Mixture of Experts (MoE) network is used to align the different node features, allowing the model to assign each node to one or more experts based on its domain and modality. 3) The aligned node features are fed into a GNN for learning and projected into a unified embedding space. 4) The decoding involves two objectives: a. Each graph domain corresponds to a specific decoder for reconstructing the node features. b. A shared shortest path distance decoder is used to reconstruct the graph structures.**

The features $x_i \in \mathbb{R}^{d_{in}}$ for node $v_i$ are then obtained by averaging the features from all modalities $\Omega_v$ associated with the node:

$$x_i = \frac{1}{|\Omega_v|} \sum_{\omega \in \Omega_v} x_i^{(\omega)}. \tag{8}$$

**Masking Node Features.** Once the features of each node are encoded, a masking strategy is applied. A subset of nodes $\tilde{\mathcal{V}} \subseteq \mathcal{V}$ is selected uniformly without replacement, and their features are replaced with a mask token $x_{[M]}$, a learnable vector $x_{[M]} \in \mathbb{R}^{d_{in}}$. This process is applied to approximately 75% of the nodes to encourage robust learning by focusing on the graph context and unmasked nodes. For each node $v_i \in \mathcal{V}$, the masked feature $\tilde{x}$ is defined as:

$$\tilde{x}_i = \begin{cases} x_{[M]} & \text{if } v_i \in \tilde{\mathcal{V}}, \\ x_i & \text{if } v_i \notin \tilde{\mathcal{V}}. \end{cases} \tag{9}$$

This masked feature $\tilde{x}$ serves as the input to the MoE, which aligns the features from different graph domains and modalities.

## 4.2 Mixture of Experts (MoE) Alignment

The MoE module [39] in UniGraph2 is designed to achieve cross-domain and cross-modality alignment by dynamically selecting specialized experts for different types of data. In MMGs, nodes may come from various domains (e.g., social networks, product networks) and have features from different modalities (e.g., text, images). A single expert network might struggle to learn appropriate representations for such diverse data. However, with the MoE architecture, the model can assign each node to one or more experts based on its domain and modality. This enables the model to adaptively align and fuse heterogeneous node features by leveraging specialized experts for specific data types. The result is a flexible

and powerful model that can learn and generalize across diverse graph structures and modalities, even when there are significant differences in feature types and distributions across domains.

Each node $v_i$ is assigned to one or more experts through a gating mechanism. Each expert $E_k$ is an MLP that processes the feature vector $\tilde{x}_i$. The final node embedding $e_i$ is computed as a weighted combination of the outputs from the selected experts:

$$e_i = \sum_{k=1}^{K} \alpha_{i,k} E_k(\tilde{x}_i). \tag{10}$$

Here, $E_k(\tilde{x}_i)$ denotes the output of expert $k$ for the node's feature vector $\tilde{x}_i$, and $\alpha_{i,k}$ represents the weight assigned to the $k$-th expert for node $v_i$. The weights $\alpha_{i,k}$ are computed using a softmax gating function, which assigns higher weights to the experts that are more relevant for the node based on its transformed features:

$$\alpha_{i,k} = \frac{\exp(g_k(\tilde{x}_i))}{\sum_{k=1}^{K} \exp(g_k(\tilde{x}_i))}, \tag{11}$$

where $g_k(\cdot)$ is the gating function that scores the relevance of expert $E_k$ for node $v_i$. The gating function $g_k$ is also an MLP that computes a scalar relevance score for each expert based on the input $\tilde{x}_i$:

$$g_k(\tilde{x}_i) = \text{MLP}_g(\tilde{x}_i)_k. \tag{12}$$

Here, the subscript $k$ denotes the $k$-th component of the gating MLP output, corresponding to the relevance score for expert $E_k$.

Thus, the MoE module adaptively routes each node's features to the most relevant experts, allowing for effective cross-domain and multimodal alignment. The experts, being specialized MLPs, capture domain-specific or modality-specific knowledge, enabling UniGraph2 to generalize well across diverse data distributions.

**GNN Encoding.** Once the aligned node embeddings $e_i$ are obtained through the MoE module, they are passed through a GNN, denoted as $f_{\text{GNN}}$, to further refine the node representations by incorporating the structural information of the graph $\mathcal{G}$. The GNN takes $e_i$ as input and propagates messages between neighboring nodes to learn the final node embeddings $h_i$:

$$h_i = f_{\text{GNN}}(e_i, \mathcal{G}). \tag{13}$$

Here, $f_{\text{GNN}}(\cdot)$ represents the GNN, which updates the embedding of each node by aggregating information from its neighbors.

**Scaling to Web-Scale Graphs.** To ensure the scalability of Uni-Graph2 on web-scale graphs, we use the Personalized PageRank (PPR) algorithm for subgraph sampling. By using PPR as the sampling strategy, we can generate the most structurally significant local subgraphs [4, 12]. Unlike other sampling methods, such as neighbor sampling or k-hop neighbors, PPR can identify key nodes and structures that hold importance in a wider context, making them more broadly applicable [19, 31].

### 4.3 Multiple Decoders

Graphs from diverse domains exhibit distinct structural and feature characteristics. A single, generic decoder would struggle to capture the specific nuances and patterns of each domain, as different types of graphs often require specialized approaches for feature reconstruction. By incorporating multiple decoders, each tailored to a specific graph domain, UniGraph2 is able to accurately reconstruct features while preserving domain-specific details.

**Feature Reconstruction.** Each decoder is responsible for reconstructing the original node features $x_i$ from the embeddings $z_i$ generated by the GNN encoder. Formally, for a domain-specific GNN decoder $f_D$, the reconstructed feature $z_i$ is obtained as:

$$z_i = f_D(h_i, \mathcal{G}). \tag{14}$$

To measure the reconstruction quality, UniGraph2 uses a cosine similarity loss [20, 22], which is defined as follows:

$$\mathcal{L}_{\text{feat}} = \frac{1}{|\tilde{\mathcal{V}}|} \sum_{v_i \in \tilde{\mathcal{V}}} \left(1 - \frac{x_i^T z_i}{\|x_i\| \cdot \|z_i\|}\right)^\gamma, \quad \gamma \geq 1, \tag{15}$$

where $x_i$ represents the original feature for node $v_i$, $z_i$ is the reconstructed feature, and $\gamma$ is a hyperparameter that controls the sharpness of the loss. This loss ensures that the reconstructed features $z_i$ maintain the same directional similarity as the original features $x_i$, encouraging accurate feature reconstruction.

**Structural Reconstruction.** In addition to reconstructing node features, UniGraph2 incorporates a shared decoder across all domains to capture structural information. Specifically, the model performs an edge-level reconstruction task to predict the shortest path distance (SPD) between node pairs, which encodes global proximity and connectivity within the graph.

The shortest path distance $\text{SPD}_{i,j}$ between nodes $v_i$ and $v_j$ is pre-computed using Dijkstra's algorithm. The loss function for shortest path distance regression is defined as:

$$\mathcal{L}_{\text{SPD}} = \frac{1}{|\mathcal{V}|^2} \sum_{(i,j) \in \mathcal{V} \times \mathcal{V}} \left\| f_{SPD}(h_i \parallel h_j) - \text{SPD}_{i,j} \right\|^2, \tag{16}$$

where $h_i$ and $h_j$ are the final GNN embeddings for nodes $v_i$ and $v_j$, respectively, $\parallel$ denotes concatenation, and $f_{SPD}$ is a task-specific head that predicts the shortest path distance between the two nodes. By regressing the SPD, the model learns to reconstruct the underlying structure of the graph, allowing it to capture the global connectivity between nodes, which is essential for tasks that depend on the graph's topology.

Then overall loss is obtained by combining the two losses with a mixing coefficient $\lambda$.

### 4.4 Inference

In the inference phase, the pre-trained UniGraph2 model is deployed to generate node embeddings for any unseen multimodal graph $\mathcal{G}^{\text{inf}} = (\mathcal{V}^{\text{inf}}, \mathcal{E}^{\text{inf}}, \mathcal{M}^{\text{inf}})$. The inference process follows a streamlined version of the training pipeline, leveraging the Modality-Specific Encoders, the MoE module, and the GNN to produce high-quality embeddings for downstream tasks such as classification, transfer learning, or generative tasks.

**Modality-Specific Encoding.** For each node $v_i \in \mathcal{V}^{\text{inf}}$, the raw features from various modalities are first processed through the respective modality-specific encoders. Let $\Omega_{v_i}^{\text{inf}} \subseteq \Omega$ represent the set of modalities associated with node $v_i$ in the inference graph. The modality-specific features are transformed as follows: $x_i^{(\omega)} = E_\omega(v_i^{(\omega)}), \quad \forall \omega \in \Omega_{v_i}^{\text{inf}}$. The node feature vector $x_i^{\text{inf}}$ is obtained by averaging the features from all available modalities: $x_i = \frac{1}{|\Omega_v|} \sum_{\omega \in \Omega_v} x_i^{(\omega)}$

**Feature Alignment.** The modality-specific feature vectors are passed through the MoE module to align and fuse information across modalities and domains. The same gating mechanism used during training is applied to select the relevant experts for each node. For each node $v_i$, the final fused embedding $e_i^{\text{inf}}$ is computed as a weighted sum of the selected experts: $e_i^{\text{inf}} = \sum_{k=1}^K \alpha_{i,k}^{\text{inf}} E_k(x_i^{\text{inf}})$, where $x_i^{\text{inf}}$ is the transformed feature of node $v_i$, and $\alpha_{i,k}^{\text{inf}}$ represents the weight assigned to expert $E_k$ for the given node, computed using the softmax gating function.

**GNN Encoding.** Once the aligned node features are obtained, they are passed through the GNN module to incorporate the structural information of the inference graph $\mathcal{G}^{\text{inf}}$. The GNN refines node embeddings by propagating messages between neighboring nodes. The output node embeddings $h_i^{\text{inf}}$ are computed as: $h_i^{\text{inf}} = f_{\text{GNN}}(e_i^{\text{inf}}, \mathcal{G}^{\text{inf}})$, where $f_{\text{GNN}}$ is the pre-trained GNN.

## 5 Experiments

In this section, we evaluate our UniGraph2 framework on three distinct research problems: 1) Self-Supervised Representation Learning, 2) Few-Shot Transfer, and 3) Multimodal Generative Tasks. Table 7 lists all 14 datasets used in the experiments.

### 5.1 Self-Supervised Representation Learning

**Setup.** We adopt the widely used linear probing protocol to evaluate the representation learning capability of self-supervised pre-trained models on unseen datasets. Specifically, we train a linear classifier on top of the embeddings generated by a frozen pre-trained model. Our model, along with all self-supervised learning baselines, is first

**Table 1: Experiment results in self-supervised representation learning. We report accuracy (%) for node/edge classification tasks and MRR (%) for link prediction tasks. UniGraph2 and other self-supervised baselines (rows in white) are jointly pre-trained on Products, Papers100M, Goodreads-LP and Amazon-Cloth, and then evaluated on the individual target dataset. *"In-distribution"* refers to pre-training on multiple datasets and evaluating on the same datasets. *"In-domain Generalization"* involves testing on target datasets from the same domain as one of the pre-training datasets. *"Out-of-domain Generalization"* evaluates on datasets from domains not seen during pre-training. The performance of methods that are directly pre-trained on the individual target dataset, is marked in gray . The methods highlighted in bold are the best-performing ones among the "rows in white" methods, while those marked in red are the best-performing methods among all methods, including those in the gray rows.**

| | In-distribution | | | | In-domain Generalization | | | | Out-of-domain Generalization | | |
|---|---|---|---|---|---|---|---|---|---|---|---|
| | Products | Papers100M | Goodreads-LP | Amazon-Cloth | Arxiv | Amazon-Sports | Goodreads-NC | Ele-fashion | Wiki-CS | FB15K237 | WN18RR |
| **Use CLIP to encode raw multimodal data as input features.** | | | | | | | | | | | |
| NoPretrain | 68.01±0.15 | 54.99±0.04 | 9.61±0.21 | 19.01±0.04 | 62.01±0.14 | 26.01±0.14 | 68.12±0.13 | 75.11±0.12 | 68.12±0.06 | 89.42±0.20 | 74.00±0.02 |
| BGRL | 70.11±0.14 | 57.12±0.05 | 20.53±0.02 | 19.11±0.01 | 65.25±0.05 | 27.35±0.05 | 72.97±0.08 | 76.53±0.02 | 70.11±0.14 | 88.11±0.12 | 73.24±0.11 |
| BGRL | 75.86±0.11 | 60.35±0.11 | 26.42±0.15 | 20.11±0.45 | 70.15±0.14 | 30.11±0.12 | 80.53±0.35 | 81.94±0.10 | 73.11±0.09 | 92.22±0.14 | 76.15±0.16 |
| GraphMAE2 | 72.25±0.16 | 60.25±0.01 | 24.11±0.14 | 19.55±0.22 | 69.18±0.02 | 28.94±0.02 | 76.18±0.05 | 77.04±0.05 | 72.15±0.14 | 90.54±0.04 | 74.11±0.13 |
| GraphMAE2 | 77.34±0.15 | 61.97±0.10 | 26.89±0.14 | 19.87±0.21 | 70.46±0.07 | 30.83±0.11 | 80.24±0.14 | 82.11±0.01 | 76.01±0.24 | 92.96±0.14 | 76.97±0.14 |
| GCOPE | 78.01±0.13 | 62.34±0.11 | 23.11±0.13 | 18.72±0.25 | 70.24±0.11 | 26.18±0.12 | 79.11±0.14 | 78.97±0.10 | 73.57±0.12 | 91.25±0.15 | 75.68±0.10 |
| **Use raw text as input features.** | | | | | | | | | | | |
| GIANT-XRT | 72.56±0.10 | 64.53±0.11 | 8.11±0.05 | 16.78±0.25 | 70.89±0.11 | 22.01±0.04 | 58.14±0.10 | 67.01±0.05 | 74.01±0.03 | 90.14±0.14 | 75.01±0.13 |
| UniGraph | 80.11±0.21 | 65.23±0.20 | 19.19±0.10 | 16.38±0.08 | 72.15±0.18 | 25.89±0.12 | 73.26±0.12 | 75.11±0.06 | 76.35±0.20 | 93.11±0.09 | 84.06±0.24 |
| UniGraph | 82.24±0.24 | 67.89±0.21 | 22.31±0.05 | 18.01±0.03 | 73.97±0.22 | 27.11±0.10 | 78.14±0.11 | 81.05±0.08 | 81.22±0.24 | 95.24±0.23 | 87.21±0.76 |
| **Use raw multimodal data as input features.** | | | | | | | | | | | |
| CLIP | 65.28±0.12 | 50.21±0.09 | 9.24±0.01 | 18.24±0.21 | 61.56±0.02 | 25.91±0.08 | 66.48±0.11 | 82.18±0.03 | 67.53±0.05 | 88.65±0.13 | 72.68±0.14 |
| ImageBind | 45.11±0.02 | 42.53±0.11 | 6.89±0.04 | 19.10±0.10 | 42.11±0.03 | 27.11±0.04 | 55.71±0.04 | 83.14±0.06 | 49.28±0.03 | 68.20±0.10 | 64.38±0.12 |
| NoPretrain | 68.34±0.14 | 55.15±0.10 | 9.62±0.02 | 19.25±0.04 | 63.76±0.11 | 25.03±0.15 | 68.01±0.15 | 83.96±0.10 | 68.45±0.10 | 89.14±0.19 | 74.01±0.15 |
| UniGraph2 | **82.79±0.02** | **67.95±0.11** | **28.98±0.11** | **24.64±0.09** | **72.56±0.15** | **30.95±0.11** | **81.15±0.12** | **85.71±0.11** | **78.15±0.09** | **94.38±0.05** | **85.47±0.11** |
| UniGraph2 | 82.36±0.21 | 67.67±0.18 | 28.76±0.08 | 24.06±0.06 | 73.46±0.17 | 31.61±0.14 | 81.97±0.10 | 87.91±0.09 | 82.86±0.07 | 95.29±0.04 | 87.86±0.06 |

jointly pre-trained on ogbn-Product, ogbn-Papers100M, Goodreads-LP, and Amazon-Cloth. We then evaluate the pre-trained models on each individual dataset. Detailed settings and hyperparameters are provided in Appendix B.

For the baselines, we compare UniGraph2 with state-of-the-art generative graph self-supervised learning methods, GraphMAE2 [20], and contrastive methods, BGRL [41]. As these methods are not inherently designed for cross-domain tasks, we leverage CLIP [37] to unify the input node features across different graphs. We also include a comparison with a multi-graph pre-training method, GCOPE [54]. UniGraph2 and all baseline methods utilize GAT [43] as the backbone GNN. For baselines that use TAGs as input, we select GIANT-XRT [55] and UniGraph [19]. Since these methods cannot process image data, they rely solely on text from MMG as node features, ignoring image inputs. For baseline approaches that accept multimodal data, we choose widely used multimodal models, CLIP [37] and ImageBind [13]. To maintain consistency with the baselines, UniGraph2 also uses CLIP's pre-trained vision and text encoders as Modality-Specific Encoders.

Our objective is to develop a general embedding model capable of generating high-quality representations for any MMG. To assess this, we evaluate the performance of UniGraph2 and the baselines in three different settings: (1) *In-distribution*, where models are pre-trained on multiple datasets and evaluated on each corresponding dataset individually; (2) *In-domain Generalization*, which tests pre-trained models on target datasets from the same domain as one of the pre-training datasets; and (3) *Out-of-domain Generalization*, where models are evaluated on datasets from domains unseen during pre-training.

**Research Questions.** In this subsection, we aim to answer the following research questions:

- **RQ1: Negative Transfer in Multi-Graph Pre-Training.** How do existing graph pre-training methods, which are primarily designed for single-graph pre-training, perform when applied to multi-graph pre-training, and how do they compare to our proposed UniGraph2?

- **RQ2: Comparison to Other Foundation Models.** How does UniGraph2, which takes both multimodal data and graph structures as input, perform compared to methods that consider only multimodal data (CLIP, ImageBind) or only TAGs (UniGraph)?

- **RQ3: Generalization Capability.** How does UniGraph2, designed as a foundation model, perform in terms of generalizing to unseen graphs, and how does it compare to methods trained directly on the target graphs?

**Results.** Table 1 presents the results. We interpret these results by answering three research questions:

- **RQ1: Negative Transfer in Multi-Graph Pre-Training.** Existing graph pre-training methods exhibit negative transfer when applied to multi-graph pre-training, whereas UniGraph2 shows improvements in this context. The results in the *In-distribution* setting demonstrate that both BGRL and GraphMAE2 experience a significant performance drop when pre-trained on multi-graphs (rows in white), compared to pre-training on single graph only (rows in gray). This suggests that pre-training on other datasets negatively affects performance on the target dataset. However, UniGraph2 shows improvement under multi-graph pre-training, indicating that it successfully addresses the shortcomings of existing graph pre-training algorithms struggling with multi-graphs.

- **RQ2: Comparison to Other Foundation Models.** UniGraph2 outperforms methods that consider only multimodal data (CLIP, ImageBind) or only TAGs (UniGraph). We observe that without

**Table 2: Experiment results in few-shot transfer. We report accuracy (%) for node/edge classification tasks. UniGraph2 and other self-supervised baselines (rows in white) are jointly pre-trained on Product, Papers100M, Goodreads-NC and Amazon-Cloth, and then evaluated on the individual target dataset. *"In-domain Generalization"* tests on target datasets from the same domain as one of the pre-training datasets. *"Out-of-domain Generalization"* evaluates on datasets from domains not seen during pre-training. The performance of methods that are direcly pre-trained on the individual target dataset, is marked in `gray`.**

| | In-domain Generalization | | | | | | | | | | | | | Out-of-domain Generalization | | | | | |
| | Cora-5-way | | PubMed-2-way | | Arxiv-5-way | | Goodreads-NC-5-way | | | Ele-fashion-5-way | | | Wiki-CS-5-way | | FB15K237-20-way | | WN18RR-5-way | |
| | 5-shot | 1-shot | 5-shot | 1-shot | 5-shot | 1-shot | 5-shot | 3-shot | 1-shot | 5-shot | 3-shot | 1-shot | 5-shot | 1-shot | 5-shot | 1-shot | 5-shot | 1-shot |
| **Use CLIP to encode raw multimodal data as input features.** | | | | | | | | | | | | | | | | | | |
| NoPretrain | 41.09 | 27.05 | 59.81 | 55.28 | 63.78 | 41.10 | 41.64 | 40.01 | 31.04 | 63.96 | 58.32 | 47.48 | 52.29 | 32.94 | 72.97 | 47.01 | 50.75 | 30.11 |
| BGRL | 52.01 | 35.18 | 66.04 | 59.04 | 60.12 | 46.67 | 47.01 | 44.22 | 30.35 | 64.72 | 60.16 | 46.49 | 52.10 | 32.85 | 75.39 | 45.15 | 47.42 | 34.57 |
| GraphMAE2 | 52.89 | 36.25 | 66.89 | 59.95 | 60.91 | 47.29 | 47.84 | 44.80 | 30.93 | 65.52 | 60.92 | 47.24 | 52.83 | 33.41 | 75.95 | 45.81 | 48.14 | 35.21 |
| Prodigy | 53.01 | 39.59 | 69.11 | 60.42 | 63.53 | 51.33 | 50.01 | 46.39 | 34.98 | 67.35 | 63.87 | 50.79 | 55.94 | 36.35 | 78.01 | 51.39 | 54.94 | 38.73 |
| OFA | 53.11 | 40.04 | 69.45 | 60.38 | 63.11 | 50.25 | 49.61 | 46.24 | 35.14 | 67.94 | 64.18 | 51.35 | 56.01 | 37.02 | 78.33 | 52.02 | 55.05 | 39.11 |
| GCOPE | 51.98 | 36.14 | 66.25 | 59.16 | 60.29 | 47.19 | 48.52 | 44.89 | 31.20 | 65.10 | 61.33 | 48.51 | 53.74 | 34.19 | 76.10 | 48.93 | 50.19 | 35.05 |
| **Use raw text as input features.** | | | | | | | | | | | | | | | | | | |
| GIANT-XRT | 50.11 | 37.85 | 68.19 | 58.78 | 62.01 | 49.01 | 46.01 | 43.86 | 30.01 | 62.97 | 61.21 | 47.76 | 54.01 | 35.04 | 76.09 | 50.25 | 53.01 | 35.19 |
| UniGraph | 54.23 | 40.45 | 70.21 | 60.19 | 64.76 | 50.63 | 46.19 | 44.01 | 33.53 | 66.21 | 62.04 | 50.17 | 56.16 | 37.19 | 78.21 | 52.19 | 55.18 | 39.18 |
| **Use raw multimodal data as input features.** | | | | | | | | | | | | | | | | | | |
| CLIP | 41.23 | 28.41 | 61.67 | 55.71 | 63.46 | 40.14 | 41.24 | 40.11 | 30.97 | 62.51 | 58.23 | 46.15 | 51.69 | 31.61 | 72.31 | 47.14 | 50.83 | 31.35 |
| ImageBind | 32.19 | 23.90 | 58.20 | 54.24 | 62.48 | 38.17 | 29.10 | 28.14 | 21.42 | 51.25 | 48.05 | 44.93 | 48.14 | 30.28 | 69.12 | 41.80 | 41.24 | 26.91 |
| NoPretrain | 42.41 | 28.39 | 60.78 | 55.90 | 64.29 | 41.98 | 42.21 | 41.20 | 31.14 | 64.15 | 58.91 | 47.90 | 52.90 | 33.14 | 74.10 | 48.11 | 51.92 | 31.84 |
| UniGraph2 | **56.01** | **42.98** | **72.19** | **61.24** | **66.24** | **51.98** | **51.73** | **47.42** | **37.01** | **69.29** | **65.29** | **53.85** | **57.28** | **38.47** | **79.34** | **52.19** | **55.59** | **39.93** |

considering the graph structure, the performance of the acknowledged powerful multimodal foundation models like CLIP is not comparable to UniGraph2. Meanwhile, UniGraph, which cannot process image data, also shows less ideal results due to the lack of information. This further highlights the necessity of designing foundation models specifically for multimodal graphs.

- **RQ3: Generalization Capability.** Compared to baseline methods, UniGraph2 demonstrates strong generalization capabilities. The results in the *In-domain Generalization* and *Out-of-domain Generalization* settings show that UniGraph2 effectively transfers knowledge from pre-training to unseen graphs. Compared to the NoPretrain method, UniGraph2 shows significant improvements. The consistent performance gains indicate that UniGraph2 can extract meaningful patterns during pre-training, which are beneficial for tackling graph learning tasks. Furthermore, UniGraph2 is comparable to methods trained directly on the target datasets, achieving similar accuracy while benefiting from greater efficiency without requiring exhaustive task-specific training.

## 5.2 Few-Shot Transfer

**Setup.** In this part, we evaluate the ability of the pre-trained models to perform few-shot in-context transfer without updating the model parameters. For baseline methods, in addition to the pre-trained models mentioned in Section 5.1, we also compare two recent graph in-context learning methods: the self-supervised pre-training method Prodigy [24] and the supervised pre-training method OFA [30].

For evaluation, we strictly follow the setting of Prodigy [24]. For an N-way K-shot task, we adopt the original train/validation/test splits in each downstream classification dataset, and construct a K-shot prompt for test nodes (or edges) from the test split by randomly selecting $K$ examples per way from the train split. By default in all experiments, we sample 500 test tasks.

We adopt the few-shot classification strategy in UniGraph [19] for UniGraph2. The model computes average embeddings for each

class and assigns a query sample to the class with the highest similarity to its embedding.

**Results.** In Table 2, our UniGraph2 model consistently outperforms all the baselines. This further demonstrates the powerful generalization capabilities of UniGraph2 as a foundation model. In particular, compared to other graph few-shot learning methods such as Prodigy, OFA, and GCOPE, UniGraph2 does not rely on complex prompt graph designs, and its simple few-shot strategy is both efficient and effective.

## 5.3 Multimodal Generative Tasks

**Setup.** UniGraph2 is designed as a general representation learning model. The embeddings it generates can be utilized by various generative foundation models, such as LLMs, to empower downstream generative tasks. To further demonstrate this, we select the section summarization task on the WikiWeb2M dataset for our experiments. The WikiWeb2M dataset [5] is designed for multimodal content understanding, using many-to-many text and image relationships from Wikipedia. It includes page titles, section titles, section text, images, and indices for each section. In this work, we focus on section summarization, where the task is to generate a summary sentence from section content using both text and images.

For the experiments, we follow the MMGL [50] setup, using four types of information: section text, section images, context text, and page-level text/images. Consistent with MMGL, we fine-tune Open Pre-trained Transformer (OPT-125m) [52] to read the input section text/images and generate a summary. Multimodal neighbors are first encoded using frozen vision/text encoders and then aligned to the text-only LM space using 1-layer MLP mapper. In MMGL, CLIP [37] encoders are used for text and image encoding, remaining frozen during fine-tuning. In our experiments, we replace CLIP embeddings with our UniGraph2 embeddings.

**Results.** Table 3 shows that under different input types and different neighbor encoding strategies, the embeddings generated by

**Table 3: Experiment results in multimodal generative tasks. We strictly follow the setting in MMGL [50]. The task is to generate a single sentence that summarizing the content of a particular section. The summary is generated based on all images and (non-summary) text present in the target and context sections. We provide different information of MMGs to the base LM: (1) section all (text + image), (2) page text, and (3) page all (all texts and images). We encode multiple multimodal neighbor information using three different neighbor encodings methods: *Self-Attention with Text+Embeddings (SA-TE)*, *Self-Attention with Embeddings (SA-E)*, and *Cross-Attention with Embeddings (CA-E)*.**

| Input Type | Method | BLEU-4 | | | | ROUGE-L | | | | CIDEr | | | |
|---|---|---|---|---|---|---|---|---|---|---|---|---|---|
| | | SA-TE | SA-E | CA-E | Avg. gain | SA-TE | SA-E | CA-E | Avg. gain | SA-TE | SA-E | CA-E | Avg. gain |
| Section all | MMGL | 8.03 | 7.56 | 8.35 | - | 40.41 | 39.89 | 39.98 | - | 77.45 | 74.33 | 75.12 | - |
| | +UniGraph2 | **9.24** | **9.01** | **9.39** | 15.57% | **43.01** | **43.24** | **42.98** | 7.44% | **81.15** | **80.39** | **81.91** | 7.32% |
| Page text | MMGL | 9.81 | 8.37 | 8.47 | - | 42.94 | 40.92 | 41.00 | | 92.71 | 80.14 | 80.72 | - |
| | +UniGraph2 | **10.31** | **10.10** | **9.98** | 14.53% | **43.19** | **43.08** | **42.75** | 3.38% | **93.19** | **90.41** | **93.11** | 9.56% |
| Page all | MMGL | 9.96 | 8.58 | 8.51 | - | 43.32 | 41.01 | 41.55 | - | 96.01 | 82.28 | 80.31 | - |
| | +UniGraph2 | **10.12** | **10.05** | **10.33** | 13.38% | **44.10** | **42.08** | **42.44** | 2.18% | **96.32** | **91.24** | **94.15** | 9.49% |

**Table 4: Ablation studies on UniGraph2 key components.**

| | Products | Amazon-Cloth | Goodreads-NC | WN18RR |
|---|---|---|---|---|
| UniGraph2 | $82.79_{\pm 0.02}$ | $\mathbf{24.64_{\pm 0.09}}$ | $\mathbf{81.15_{\pm 0.12}}$ | $\mathbf{85.47_{\pm 0.11}}$ |
| w/o MoE | $81.01_{\pm 0.10}$ | $21.33_{\pm 0.04}$ | $80.10_{\pm 0.04}$ | $83.99_{\pm 0.21}$ |
| w/o feat loss | $69.12_{\pm 0.09}$ | $18.43_{\pm 0.24}$ | $68.12_{\pm 0.01}$ | $74.11_{\pm 0.03}$ |
| w/o SPD loss | $82.42_{\pm 0.11}$ | $23.39_{\pm 0.05}$ | $80.24_{\pm 0.02}$ | $85.24_{\pm 0.11}$ |

**Table 5: Ablation studies on Modality-Specific Encoders.**

| | Products | Amazon-Cloth | Goodreads-NC | WN18RR |
|---|---|---|---|---|
| CLIP | $82.79_{\pm 0.02}$ | $24.64_{\pm 0.09}$ | $81.15_{\pm 0.12}$ | $\mathbf{85.47_{\pm 0.11}}$ |
| ImageBind | $82.32_{\pm 0.05}$ | $\mathbf{25.01_{\pm 0.11}}$ | $80.33_{\pm 0.22}$ | $84.29_{\pm 0.07}$ |
| T5+ViT | $\mathbf{82.99_{\pm 0.04}}$ | $24.38_{\pm 0.28}$ | $\mathbf{81.28_{\pm 0.11}}$ | $84.16_{\pm 0.04}$ |

**Table 6: Comparison of GPU hours and performance on ogbn-Arxiv and ogbn-Papers100M.**

| Method | Pre-training | Downstream Training | Downstream Inference | Test Accuracy |
|---|---|---|---|---|
| **ogbn-Arxiv (169,343 nodes)** | | | | |
| GAT | - | 0.39 h | 5.5 mins | 70.89 ± 0.43 |
| GraphMAE2 | - | 5.1 h | 5.4 mins | 70.46 ± 0.07 |
| UniGraph | 28.1 h | - | 9.8 mins | 72.15 ± 0.18 |
| UniGraph2 | 5.2 h | - | 5.7 mins | **72.56 ± 0.15** |
| **ogbn-Papers100M (111,059,956 nodes)** | | | | |
| GAT | - | 6.8 h | 23.1 mins | 65.98 ± 0.23 |
| GraphMAE2 | - | 23.2 h | 23.0 mins | 61.97 ± 0.24 |
| UniGraph | 28.1 h | - | 40.1 mins | 67.89 ± 0.21 |
| UniGraph2 | 5.2 h | - | 24.8 mins | **67.95 ± 0.11** |

UniGraph2 bring significant improvements compared to MMGL's default CLIP embeddings. We also observe that UniGraph2's embeddings are more robust to different neighbor encoding strategies compared to CLIP and do not rely on a specific strategy.

## 5.4 Model Analysis

We select four datasets from different domains to conduct more in-depth studies. We adopt self-supervised representation learning for evaluation.

**Ablation on Key Components.** Table 4 shows the performance of the UniGraph2 framework after removing some key designs. "W/o MoE" represents that we use simple MLP instead MoE to align node features. "W/o feat loss" represents that we only use the SPD loss

for pre-training, while "w/o SPD loss" refers to the opposite. The overall results confirm that all key designs contribute positively to the performance of UniGraph2.

**Ablation on Modality-Specific Encoders** In Table 5, we study the influence of different Modality-Specific Encoders on the performance of encoding raw multimodal data. CLIP and ImageBind are feature encoders that map features from various modalities to a shared embedding space, whereas T5+ViT employs SOTA embedding methods for each modality independently, without specific alignment. The results show that all methods achieve comparable performance, indicating that UniGraph2 effectively aligns features regardless of whether they have been pre-aligned or not.

**Efficiency Analysis.** UniGraph2, designed as a foundation model, incurs significant computational costs primarily during the pre-training phase. However, it offers the advantage of applicability to new datasets in the inference phase without requiring retraining. We compare of the training and inference costs of our model with other models. GAT [43] is a supervised trained GNN. GraphMAE2 [20] is a self-supervised learning method with GAT as the backbone network. UniGraph [19] is a graph foundation model for TAGs. We select ogbn-Arxiv and ogbn-Papers100M, two datasets of different scales for experiments. From the results in the Table 6, we observe that although UniGraph2 has a long pre-training time, its inference time on downstream datasets is comparable or shorter than the combined training and inference time of GNN-based methods. This advantage further increases with the size and potential quantity of downstream datasets.

## 6 Conclusion

UniGraph2 addresses the limitations of existing foundation models for multimodal graphs by introducing a novel unified embedding space that effectively integrates both multimodal information and graph structures. By employing modality-specific encoders, a graph neural network, and a Mixture of Experts module, UniGraph2 outperforms state-of-the-art models in tasks such as classification, transfer learning, and multimodal generation. Extensive experiments demonstrate the model's generalization capabilities across diverse graph domains and modalities, confirming its potential as a scalable and flexible solution for learning on multimodal graphs.

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

## A Datasets

**Cora** [16]. The Cora dataset consists of 2708 scientific publications classified into one of seven classes – case based, genetic algorithms, neural networks, probabilistic methods, reinforcement learning, rule learning, and theory. The citation network consists of 5429 links. We collect raw text from [16].

**PubMed** [16]. The Pubmed dataset consists of 19,717 scientific publications from PubMed database pertaining to diabetes classified into one of three classes – Experimental induced diabetes, Type 1 diabetes, and Type 2 diabetes. As in [29], we ask ChatGPT to generate a detailed description of each category. The citation network consists of 44,338 links. We collect raw text from [16].

**ogbn-Arxiv** [23]. The ogbn-arxiv dataset is a directed graph, representing the citation network between all Computer Science (CS) arXiv papers. Each node is an arXiv paper and each directed edge indicates that one paper cites another one. The task is to predict the 40 subject areas of arXiv CS papers, e.g., cs.AI, cs.LG, and cs.OS. We collect raw text from [23].

**ogbn-Papers100M** [23]. The ogbn-papers100M dataset is a directed citation graph of 111 million papers. We collect raw text from [23].

**ogbn-Products** [23]. The ogbn-products dataset is an undirected and unweighted graph, representing an Amazon product co-purchasing network. Nodes represent products sold in Amazon, and edges between two products indicate that the products are purchased together. The task is to predict the category of a product in a multi-class classification setup, where the 47 top-level categories are used for target labels. We collect raw text from [23].

**Wiki-CS** [29]. Wiki-CS is a Internet link network with each node represent a Wikipedia page and each edge represent the reference link. Each node's label corresponds to the category of the entry. We collect raw text from [29].

**FB15K237** [29]. FB15K237 is a kowledge graph that contains knowledge base relation triples and textual mentions of Freebase entity pairs. We collect raw text from [29]. Given that we propose a self-supervised learning framework, and the edge text features are the labels to be predicted, we solely utilized node text features and did not employ edge text features.

**WN18RR** [29]. WN18RR is a knowledge graph, which is a subset of WordNet that consists of 11 relations and 40943 entities. We collect raw text from [29]. Given that we propose a self-supervised learning framework, and the edge text features are the labels to be predicted, we solely utilized node text features and did not employ edge text features.

**Amazon-Sports** [56]. Amazon-Sports is a link prediction dataset derived from the Amazon-Review dataset. In this dataset, each node represents a product within the sports category on Amazon, and the links signify whether two products are often purchased together. The textual features consist of product titles, while the visual features are raw high-resolution images of the products. We collect raw text and images from [56].

**Amazon-Cloth** [56]. Amazon-Cloth follows a similar structure to Amazon-Sports, but focuses on clothing products. The dataset uses co-purchase information from the clothes category on Amazon. The text features include product titles, such as "Nike Men's Revolution 6 Road Running," and the visual features are the associated product images. We collect raw text and images from [56].

**Goodreads-LP** [56]. Goodreads-LP is based on the Goodreads Book Graph dataset. In this dataset, nodes correspond to books, and the links represent whether users who like one book are likely to enjoy another. Text features describe the books, and the visual features are book cover images. Books without images are excluded from the dataset. We collect raw text and images from [56].

**Goodreads-NC** [56]. Goodreads-NC is a node classification dataset also based on the Goodreads dataset. Here, each node represents a book, and the links signify whether users who liked one book will like another. The textual features describe the books, and the visual features are book cover images. Books lacking images are removed. We collect raw text and images from [56].

**Ele-Fashion** [56]. Ele-Fashion is a node classification dataset derived from the Amazon-Fashion dataset. In this dataset, each node represents a fashion product, and links indicate that users who buy one product are likely to purchase another. The textual features are product titles, and the visual features consist of product images. We collect raw text and images from [56].

**WikiWeb2M** [5]. The WikiWeb2M dataset is designed for multimodal content understanding, using many-to-many text and image relationships from Wikipedia. It includes page titles, section titles, section text, images, and indices for each section.

## B Implementation Notes

**Running environment.** All experiments are conducted on Linux machine with 945G RAM, and 8 NVIDIA A100 with 40GB GPU memory. For software versions, we use Python 3.11, Pytorch 2.0.1, DGL 1.1.2, transformers 4.32.1 and CUDA 11.8. Our code and datasets will be available.

**Table 7: Statistics of all 14 multimodal graph datasets.**

| Dataset | Domain | Task | #Nodes | #Edges | Raw Features |
|---|---|---|---|---|---|
| Cora | Citation | Node | 2,708 | 5,429 | Paper Titles and Abstracts |
| PubMed | Citation | Node | 19,717 | 44,338 | Paper Titles and Abstracts |
| ogbn-Arxiv | Citation | Node | 169,343 | 1,166,243 | Paper Titles and Abstracts |
| ogbn-Papers100M | Citation | Node | 111,059,956 | 1,615,685,872 | Paper Titles and Abstracts |
| ogbn-Products | Product | Node | 2,449,029 | 61,859,140 | Product Descriptions |
| Wiki-CS | Wikipedia | Node | 11,701 | 216,123 | Wikipedia Entry Names and Contents |
| Ele-fashion | Product | Node | 97,766 | 199,602 | Fashion Titles and Fashion Images |
| Goodreads-NC | Book | Node | 685,294 | 7,235,084 | Book Descriptions and Book Images |
| FB15K237 | Knowledge | Edge | 14,541 | 310,116 | Entity Names and Descriptions |
| WN18RR | Knowledge | Edge | 40,943 | 93,003 | Entity Names and Descriptions |
| Amazon-Sports | Product | Edge | 50,250 | 356,202 | Product Titles and Product Images |
| Amazon-Cloth | Product | Edge | 125,839 | 951,271 | Product Titles and Product Images |
| Goodreads-LP | Book | Edge | 636,502 | 3,437,017 | Book Descriptions and Book Images |
| WikiWeb2M | Wikipedia | Generative | 600,000 | - | Page Title, Section Titles, Section Text, Images |

**Table 8: Notation Table**

| Symbol | Description |
|---|---|
| $\mathcal{G} = (\mathcal{V}, \mathcal{E}, \mathcal{M}, \Omega)$ | A Multimodal Graph (MMG). |
| $\mathcal{V}$ | Set of nodes in the graph. |
| $\mathcal{E}$ | Set of edges in the graph. |
| $\Omega$ | Set of possible modalities (e.g., text, images). |
| $\mathcal{M}(v)$ | Function that maps each node $v \in \mathcal{V}$ to a subset of modalities $\Omega_v \subseteq \Omega$. |
| $\mathcal{G}_{\text{TAG}} = (\mathcal{V}, \mathcal{E}, \mathcal{M}, \{\text{text}\})$ | Text-attributed graph where each node has an associated text feature. |
| $f : \mathcal{V}_k \rightarrow \mathbb{R}^d$ | Pre-trained model for representation learning, mapping nodes to a $d$-dimensional embedding space. |
| $H_{\text{inf}}$ | Inference embeddings generated by applying the pre-trained model to a new graph. |
| $x_i^{(\omega)}$ | Feature vector for node $v_i$ from modality $\omega$. |
| $\mathcal{G}_{\text{inf}} = (\mathcal{V}_{\text{inf}}, \mathcal{E}_{\text{inf}}, \mathcal{M}_{\text{inf}})$ | Inference graph where the pre-trained model generates embeddings for nodes. |
| $\mathcal{L}_{\text{feature}}$ | Feature reconstruction loss for reconstructing masked node features. |
| $\mathcal{L}_{\text{SPD}}$ | Shortest path distance reconstruction loss used for structural reconstruction. |
| $\lambda$ | Mixing coefficient for combining feature and structure reconstruction losses. |

**Hyper-parameters.** The detailed pre-training hyper-parameters are listed in Table 9. For linear probing, we train the linear classifier using adam optimizer with lr=0.01 for 5000 epochs, and report the early-stopping results.

**Baselines.** To have a fair comparison, we download the public source code. For methods can not scale, we adapt their code to integrate with sampling algorithms to run on large-scale graphs. The sources of the codes used are as follows:

- BRGL: https://github.com/Namkyeong/BGRL_Pytorch
- GraphMAE2: https://github.com/THUDM/GraphMAE2
- GIANT-XRT: https://github.com/amzn/pecos/tree/mainline/examples/giant-xrt
- Prodigy: https://github.com/snap-stanford/prodigy
- OFA: https://github.com/LechengKong/OneForAll
- UniGraph: https://github.com/yf-he/UniGraph
- CLIP: https://github.com/openai/CLIP
- ImageBind: https://github.com/facebookresearch/ImageBind
- GCOPE: https://github.com/cshhzhao/gcope

- MMGL: https://github.com/minjiyoon/MMGL

**Datasets splits.** For Cora and PubMed, we follow commonly used data splits, using 20 labeled nodes per class as the training set, 30 nodes per class as the validation set, and the rest as the test set. We report the average accuracy on test set with 20 random initialization.

For Arxiv and Products, we follow the official splits [23]. Following the experimental procedure suggested by OGB, we repeat each experiment for 10 times with random seeds and report the average accuracy.

For Wiki-CS, we follow the official splits [32] with 20 different training splits, we report the average accuracy on the 20 different training splits with 20 random initialization. In each split, 5% of the nodes in each class are used for training.

For FB15K237 and WN18RR, we follow splits in OFA [29]. For FB15K237, training set has 272115 edges, validation set has 17535 edges and test set has 20466 edges. For WN18RR, training set has 86835 edges, validation set has 3034 edges and test set has 3134

**Table 9: Pre-training hyper-parameters for our framework.**

| mask rate | hidden_size | lr | weight_decay | dropout | optimizer | num_epochs | num_gnn_layers | ppr topk | num_experts | coefficient $\lambda$ |
|---|---|---|---|---|---|---|---|---|---|---|
| 0.8 | 1024 | 1e-3 | 0.01 | 0.4 | adamw | 5 | 4 | 256 | 8 | 0.1 |

edges. We repeat each experiment for 10 times with random seeds and report the average accuracy.

For Amazon-Sports, Amazon-Cloth, Goodreads-LP, Goodreads-NC, and Ele-Fashion, we follow the official splits [56]. We repeat each experiment for 10 times with random seeds and report the average accuracy.

For WikiWeb2M, we follow the split and setting in MMGL [50].

**Linear probing.** The dataset $\mathcal{D}$ after generating embeddings, comprising embedding-label pairs $(\boldsymbol{h}, y)$, is divided into training, validation, and test sets. A linear classifier with weight matrix $\boldsymbol{W} \in \mathbb{R}^{d \times |\mathcal{Y}|}$ is trained at top the embeddings from the frozen model, aiming to minimize the loss function $\mathcal{L}$, typically cross-entropy, over the training set: $\min_{\boldsymbol{W}} \sum_{(\boldsymbol{h}, y) \in \mathcal{D}_{\text{train}}} \mathcal{L}(\boldsymbol{W} \cdot \boldsymbol{h}, y)$. The performance of the model is evaluated based on a performance metric $\mathcal{M}$, which can be defined generically as $\mathcal{M}(\mathcal{D}_{\text{eval}}, f_\theta, \boldsymbol{W})$, where $\mathcal{D}_{\text{eval}}$ refers to either the validation or test set.

**Few-shot transfer.** Our method follows in-context learning approach in UniGraph [19], and for baselines we either follow the same approach or use their already proposed in-context learning methods (Prodigy, OFA). We repeat each experiment for 10 times with random seeds and report the average accuracy. All the other experimental details (pre-training) follow those for the previous experiment (i.e., linear probing).

## C Mixture of Experts (MoE) in Graph Learning

Mixture of Experts (MoE) is a machine learning architecture that distributes the learning task across several specialized expert models. In various implementations of MoE in graph neural networks (GNNs), each expert model is typically responsible for learning specific components of the data or task, and a gating model selects which expert(s) to activate for each input, effectively combining their outputs. As in MoE in NLP, most MoE in graph learning are designed to improve efficiency in inference [45]. Other works also use MoE to handle different challenges like distribution shifts. In Graph-METRO [48], MoE addresses complex graph distribution shifts by assigning each expert to deal with a specific shift type, while a gating model selects the relevant experts to produce shift-invariant representations. GraphAlign [21] uses a feature normalization step and employs MoE at the input layer to assign nodes to experts, ensuring a unified distribution across graphs before GNN training. In this work, UniGraph2 employs MoE to align multimodal features (e.g., text, images) from various graph domains, ensuring coherent embeddings across modalities and domains.

