# OpenReview forum: "UniGraph2: Learning a Unified Embedding Space to Bind Multimodal Graphs"
_ACM.org/TheWebConf/2025/Conference — WWW 2025 Oral_

### Official Review · Reviewer_nzEj · 2024-11-01

**Novelty:** 4
**Technical Quality:** 6

**Review:**

Paper summary:
This paper introduces UniGraph2, a foundation model for multimodal graphs (MMGs) that incorporates both entity relationships and multimodal data, addressing limitations in existing models like CLIP, which lack graph structure modeling. UniGraph2 combines modality-specific encoders with a GNN to create unified embeddings that capture multimodal and relational information. A cross-domain multi-graph pre-training algorithm and a Mixture of Experts (MoE) component further enhance transferability and coherence across domains and modalities. Experiments demonstrate UniGraph2's superior performance across MMG tasks.

Pros:
1. This work is valuable for enhancing performance on MMG datasets through an effective, unified framework.
2. The framework’s applicability extends beyond MMGs to multimodal generative tasks, showcasing its versatility.
3. The design components, especially the novel MoE module, are intuitive and innovative.

Cons:
1. The work appears incremental, building on UniGraph with additions like the MoE component and a novel loss function.
2. The motivation for Structural Reconstruction in Equation 16 could be clearer, and additional references would strengthen this aspect.
3. The study lacks comparisons with related baselines, such as the MMGCN model for multimodal applications [1].

[1] Hu, Jingwen, et al. "MMGCN: Multimodal fusion via deep graph convolution network for emotion recognition in conversation." arXiv preprint arXiv:2107.06779 (2021). https://arxiv.org/abs/2107.06779

**Questions:**

In Table 3, the authors present various neighbor encoding methods: Self-Attention with Text+Embeddings (SA-TE), Self-Attention with Embeddings (SA-E), and Cross-Attention with Embeddings (CA-E). Could the authors provide more details on these neighbor encoding methods?

**Reviewer Confidence:**

3: The reviewer is confident but not certain that the evaluation is correct

**Scope:**

4: The work is relevant to the Web and to the track, and is of broad interest to the community

---

### Official Review · Reviewer_PebM · 2024-11-10

**Novelty:** 4
**Technical Quality:** 6

**Review:**

This paper presents a multimodal graph foundation model that unifies features from different modalities into a single embedding space. The model integrates modality-specific encoders, a GNN, and a MoE module. It is evaluated on various tasks, including self-supervised learning, few-shot transfer, and generative tasks.

**Pros:**
- The paper is technically sound.
- The experiments are extensive, and the results support the model’s design.
- The code is accessible.
- Using different encoders to learn distinct modalities and mapping these to a unified space is interesting.

**Cons:**
- The main concern is that the model combines numerous components, making it appear overly complex. Many components seem derived from existing methods, so I encourage the authors to provide a clearer rationale for the necessity of each component.

**Questions:**

1. If there is a new modality, does this modality only need to train the modality-specific encoder?


2. Is there a visualization to show how different modalities in the unified space?

3. Do you have ablation analysis of the weights for $L_{feat} $ and $ L_{SPD} $?

**Reviewer Confidence:**

2: The reviewer is willing to defend the evaluation, but it is likely that the reviewer did not understand parts of the paper

**Scope:**

4: The work is relevant to the Web and to the track, and is of broad interest to the community

---

### Official Review · Reviewer_ghGf · 2024-12-01

**Novelty:** 5
**Technical Quality:** 5

**Review:**

UniGraph2 is a cross-domain graph foundation model designed to address the challenges in learning graph structures in multimodal graphs (MMGs). Unlike traditional multimodal embedding models, UniGraph2 combines modality-specific encoders with a graph neural network (GNN) to capture both multimodal information and graph structure through a low-dimensional embedding space. It introduces a cross-domain multi-graph pre-training algorithm and a Mixture of Experts (MoE) component to enhance feature alignment and transfer learning. Experiments demonstrate that UniGraph2 outperforms existing models in representation learning, transfer learning, and multimodal generative tasks, offering a scalable and flexible solution for MMG learning.

- pros

(1) The paper is well-organized and presents a multimodal graph foundation model.

(2) The experiments are thorough, effectively demonstrating the model's validity.

(3) The model is highly scalable and can serve as a reference for other tasks.

- cons

(1) MOE dynamically assigns weights to each expert, but I do not understand how this can achieve cross-domain and cross-modal alignment. The rationale for using Mixture of Experts (MoE) is somewhat weak.

(2) The specific meaning of each expert is unclear; it currently operates as a complete black box.

**Questions:**

(1) The authors should provide a more reasonable explanation for how using MOE achieves alignment.

(2) The visualization of each MOE expert should be presented and analyzed in the experiments.

**Reviewer Confidence:**

3: The reviewer is confident but not certain that the evaluation is correct

**Scope:**

4: The work is relevant to the Web and to the track, and is of broad interest to the community

---

### Official Review · Reviewer_BpYj · 2024-12-02

**Novelty:** 4
**Technical Quality:** 4

**Review:**

This paper proposes a universal multimodal graph representation model applicable to cross-modal and cross-domain graph scenarios. To achieve this, the authors first introduce a representation model comprising three components: 1) a multimodal encoder, which extracts features for data from different modalities through respective encoders; 2) a mixture-of-experts (MoE) module designed to better align different modalities; and 3) a graph neural network (GNN) for modeling based on the graph's topological structure. Subsequently, the authors propose a pre-training method to train the model by reconstructing node and edge information from the representation. Finally, the authors validate the effectiveness of the proposed model through experiments and verify the contribution of each component via ablation studies.


1. Compared to previous work on graph representation, the proposed method can simultaneously extract representations for graph data from different modalities and domains.
2. The authors demonstrate the effectiveness of the proposed method through three tasks (Self-Supervised Representation Learning, Few-Shot Transfer, and Multimodal Generative Tasks). Additionally, the components of the method are validated through ablation experiments.

**Questions:**

1. The explanation of "Masking Node Features" (line 380) is unclear:
   1.1. Does the 75% mentioned in line 384 refer to the proportion of nodes masked or the proportion of nodes not masked?
   1.2. For subsequent modules in the method, how are different masked nodes distinguished?
   1.3. How was the 75% value selected? Are there ablation studies for this part?
2. In the related work section, MMGL [1] is mentioned. Although this work targets generative tasks, it should also be capable of extracting graph node representations. Can the performance of representations obtained from this model be compared with your method?
3. The ablation study of the MoE module is not described clearly. When replacing the MoE module with a standard MLP, were other variables, such as the total number of model parameters, kept consistent?
4. Minor typos, such as the issue with the graph notation in line 339.

[1] Yoon, Minji, et al. "Multimodal graph learning for generative tasks." arXiv preprint arXiv:2310.07478 (2023).

**Reviewer Confidence:**

3: The reviewer is confident but not certain that the evaluation is correct

**Scope:**

3: The work is somewhat relevant to the Web and to the track, and is of narrow interest to a sub-community

---

### Official Review · Reviewer_DTL2 · 2024-12-06

**Novelty:** 5
**Technical Quality:** 5

**Review:**

This paper introduces UniGraph2, a graph foundation model that establishes a unified embedding space integrating both data graph structures and multimodal information. UniGraph2 is designed for broad applicability across a variety of graph domains and tasks. It utilizes modality-specific encoders, the Mixture of Experts (MoE) module, and decoders to enhance its processing capabilities. Several experiments have been conducted to show the efficacy of the proposed UniGraph2 across various multimodal graph learning tasks.

Pros:

1. The paper is clearly written and easy to follow.

2. The exploration of a unified embedding for both graph structures and multimodal information that can be generalized across various domains addresses a crucial and challenging area in the field.

3. The integration of a dynamic Mixture of Experts module, followed by GNN encoding, is interesting and provides the model with great representation capacity.

4. This paper evaluates the performances across a range of multimodal graph learning tasks and datasets.

Cons:

1. Decoder Training Overhead: UniGraph2 necessitates the training of distinct domain-specific decoders for each new domain, which introduces additional overhead.

2. Clarity of Evaluation Metrics: The presentation of evaluation metrics in Tables 3, 4, and 5 lacks clarity.

3. Formatting Issue: The document contains an overfull error at line 548.

**Questions:**

1. How does the number of experts in the Mixture of Experts component, as well as the presence of the gating
mechanism, influence the representation performance of UniGraph2?

2. In Table 6, UniGraph2 shows a significantly shorter pre-training time compared to the UniGraph model. Could
you provide specific reasons?

3. Are there specific domains or types of multimodal data where UniGraph2 fails to perform as expected?

4. Could the authors extend the evaluation of UniGraph2 to include Retrieval Tasks? It would be insightful to
see how the model performs on such tasks compared to existing benchmarks.

**Reviewer Confidence:**

3: The reviewer is confident but not certain that the evaluation is correct

**Scope:**

3: The work is somewhat relevant to the Web and to the track, and is of narrow interest to a sub-community